# Loss-of-function cancer-linked mutations in the EIF4G2 non-canonical translation initiation factor

Sara Meril[1], Marcela Bahlsen[1], Miriam Eisenstein[1] [iD], Alon Savidor[2] [iD], Yishai Levin[2], Shani Bialik[1], Shmuel Pietrokovski[1] [iD], Adi Kimchi[1] [iD]

**Tumor cells often exploit the protein translation machinery, resulting in enhanced protein expression essential for tumor growth. Since canonical translation initiation is often suppressed because of cell stress in the tumor microenvironment, non-canonical translation initiation mechanisms become particularly important for shaping the tumor proteome. EIF4G2 is a non-canonical translation initiation factor that mediates internal ribosome entry site (IRES)- and uORF-dependent initiation mechanisms, which can be used to modulate protein expression in cancer. Here, we explored the contribution of EIF4G2 to cancer by screening the COSMIC database for EIF4G2 somatic mutations in cancer patients. Functional examination of missense mutations revealed deleterious effects on EIF4G2 protein–protein interactions and, importantly, on its ability to mediate non-canonical translation initiation. Specifically, one mutation, R178Q, led to reductions in protein expression and near-complete loss of function. Two other mutations within the MIF4G domain specifically affected EIF4G2's ability to mediate IRES-dependent translation initiation but not that of target mRNAs with uORFs. These results shed light on both the structure–function of EIF4G2 and its potential tumor suppressor effects.**

## Introduction

Eukaryotic mRNA translation is a highly regulated multistep process that regulates protein synthesis under normal and stress conditions. mRNA translation is largely divided into three steps: initiation, elongation, and termination (1). In canonical cap-dependent translation, the initiation step involves the recruitment of eukaryotic initiation factors of the multiprotein complex EIF4F on the mRNA 5′ cap m7GTP structure (m7G cap). EIF4F is comprised of the cap binding EIF4E, the helicase EIF4A, and EIF4G1, which acts as a scaffold bridging the aforementioned factors, as well as PABP and EIF3, to enable 40S ribosome recruitment (1). Notably,

non-canonical mechanisms of translation initiation have also been described. One of the main players in these non-canonical mechanisms is EIF4G2 (also known as DAP5/p97/Nat1), a member of the EIF4G family of initiation factors (2, 3, 4). Unlike EIF4G1, EIF4G2 lacks the PABP and EIF4E binding sites, and therefore instead participates in several cap-independent modes of translation initiation, such as those using internal ribosome entry sites (IRESes) (5, 6, 7, 8, 9, 10) or cap-independent translation enhancers (11), and N6-methyladenosine –driven translation (12). In addition, EIF4G2 has been shown to promote read-through of 5′ uORFs (13) and/or re-initiation of the main coding sequence (CDS) after cap-dependent translation of these uORFs (14). It can also mediate a non-canonical cap-dependent method of initiation by binding EIF3D (15, 16), which replaces EIF4E as the mRNA 5′ cap interactor (17).

In order to maintain enhanced proliferation and altered cellular metabolism, tumors need to up-regulate their protein translation capacity, and as such, cap-dependent translation becomes a convergent point for regulation in cancer cells (18). In fact, the involvement of the canonical initiation factors in cancer formation and progression is well established (19). For example, EIF4E is often highly expressed in different cancers and can drive the translation of specific mRNAs such as VEGFA, MYC, and TGFB1, which ultimately promote angiogenesis, cell proliferation, and oncogenesis. Similar functions and cancer-promoting outcomes have been shown for the overexpression of additional translation initiation factors such as EIF4A and EIF4G1 (19). Yet, the tumor microenvironment is often characterized by cell stress, during which canonical cap-dependent translation is suppressed. In these circumstances, non-canonical translation initiation mechanisms, such as those involving IRESes and uORFs, are used (18, 19). Yet although these specific mechanisms have been shown to be important for specific oncogene expression, such as MYC, little is known about how non-canonical translation initiation factors driving these mechanisms may contribute to cancer development and progression.

EIF4G2 is an appealing candidate to be studied in the context of cancer because of its established physiological roles. EIF4G2 is critical for embryonic development and differentiation of

---

[1]Department of Molecular Genetics, Weizmann Institute of Science, Rehovot, Israel    [2]The de Botton Institute for Protein Profiling of the Nancy and Stephen Grand Israel National Center for Personalized Medicine (G-INCPM), Weizmann Institute of Science, Rehovot, Israel

Correspondence: adi.kimchi@weizmann.ac.il

---

embryonic stem cells by driving non-canonical selective translation of critical mRNA cohorts (20, 21, 22). In addition, EIF4G2's established translation targets during cell stress and apoptosis, including pro-apoptotic proteins APAF1 and MYC, and anti-apoptotic IAP proteins (5, 10, 23, 24), and BCL2, BCL2L1, and CDK1 during mitosis (25, 26), indicate its involvement in cell death and survival pathways. It is also involved in cellular responses to hypoxia and stress in various cancer cells, by mediating translation of PHD2 (27) and an NH2-terminal truncated TP53 isoform from an internal IRES (28), respectively. These specific functions suggest that EIF4G2 may be critical for cell fate decisions in cancer, although its multiple and sometimes opposing functions make it difficult to predict whether EIF4G2 can promote or suppress tumor development and growth. To date, only a small number of studies have examined a direct role of EIF4G2 in cancer. High *EIF4G2* mRNA expression was associated with gastric cancer and metastatic triple-negative breast cancer, correlating with decreased overall and metastasis-free survival (15, 29). In metastatic breast cancer cells, knock-down (KD) of EIF4G2 resulted in decreased cell migration in cellular invasion and wound healing assays, and increased apoptosis upon loss of cell adherence. Injection of the KD cells into mice produced tumors with similar growth as control cells, but with decreased metastasis, invasiveness, and angiogenesis (30). Consistent with these functional data, many of the EIF4G2 target mRNAs in metastatic breast cancer cells were associated with cell migration, invasion, the epithelial-to-mesenchymal transition, and survival, such as integrins, vimentin, SNAIL1/2, and ZEB1 (30). It is thus likely that in breast cancer, EIF4G2 promotes metastasis through its ability to enhance migration, invasion, and cell survival. On the contrary, EIF4G2 expression was observed to be reduced in bladder cancer, correlating with tumor dedifferentiation and invasiveness (31). Thus, the limited data to date suggest that EIF4G2 may act as either an oncogene or a tumor suppressor, depending on the tumor context, and call for a broader and deeper investigation into whether EIF4G2 gain or loss of function is indeed associated with cancer development and/or progression.

Here, we screened the COSMIC database for *EIF4G2* somatic mutations in cancer patients. We focused on the possible effects of single missense mutations on EIF4G2 protein structure and function, specifically on protein–protein interactions and translation initiation functions. Through analysis of these mutations, we have established the occurrence of loss-of-function mutations in EIF4G2 and separated the IRES-dependent initiation functions from uORF-dependent initiation functions, opening the door to understanding the phenotypic outcome of its loss of function on cancer progression and aggressiveness.

## Results and Discussion

### Screening for EIF4G2 mRNA expression and mutational burden in cancer patients

TCGA database was screened in an unbiased manner for *EIF4G2* mRNA expression in healthy subjects compared with patients harboring primary tumors (Fig S1A and B). The analysis was performed on 24 different cancer histology subtypes for which healthy subjects' data were also available, taking into consideration that the healthy and primary tumor specimens were not necessarily sequenced from the same individual. Notably, nearly one third of the analyzed subtypes showed a significant reduction in *EIF4G2* mRNA expression (i.e., uterine corpus endometrial carcinoma, bladder urothelial carcinoma, kidney renal clear cell carcinoma, prostate adenocarcinoma, head-and-neck squamous cell carcinoma, kidney renal papillary cell carcinoma, and thyroid carcinoma; 7 of 24 analyzed tumors), whereas two tumors (glioblastoma multiforme and cholangiocarcinoma) demonstrated a significant increase in *EIF4G2* mRNA expression. Overall, these results demonstrate that *EIF4G2* mRNA expression differs according to the tumor type.

In light of the inconclusive nature of the expression data, and considering that EIF4G2 is a long-lived protein under specific translation control (32, 33, 34), whereby its mRNA expression is particularly limited in its ability to predict EIF4G2 protein expression, we sought to determine a more definitive indicator of the EIF4G2 function in patient tumors. In the absence of publicly available informative, abundant data on protein expression, we instead examined somatic mutations derived from cancer patients to determine whether they serve as gain-of-function or loss-of-function mutations, or affect the protein levels of EIF4G2.

Sequencing data of 411 *EIF4G2* mutations from 369 different patients, all confirmed to be independent somatic mutations, were screened and collected from the COSMIC database. Of these 369 samples, 290 (79%) were carcinomas and 42 (11%) were melanomas, with the remainder being cancers of different histology types at low percentages (Fig 1A). Focusing on carcinoma, the most abundant histology type, 101 (~35%) mutations were found in adenocarcinomas, 32 (11%) in squamous cell carcinomas, and 21 (7%) in endometrioid carcinomas (Fig 1A). 248 of the mutations were situated in the CDS, 77% (191/248) of which were categorized as missense mutations (Fig 1B). Using data analysis and statistical methods that carefully identify primary unique mutations and calculate their significance using the expected probabilities of all possible mutations (35), six hotspots (26/191, 13.6%) with a statistically significant occurrence were identified (Fig 1C). In addition, 10.5% (26/248) of the mutations in the *EIF4G2* CDS predicted a complete loss of function because of out-of-frame deletion, insertion, or early stop codon (nonsense) mutations (Fig 1B). The distribution across cancer types of the predicted deleterious mutations and the significant missense mutations is shown in Fig 1D, and represents the number of patients with each annotated mutation according to the primary tumor site. This in-depth, accurate statistical analysis of numerous naturally occurring tumor primary sites revealed that *EIF4G2* mutations are mostly found in cancers of the large intestine and endometrium, representing 22% and 14%, respectively, out of the deleterious and significant missense mutations presented. These results emphasize the possible biological importance of EIF4G2 in colon and endometrial tumors. Moreover, the complete loss-of-function mutations are an important indication that the EIF4G2 protein is probably reduced in both quantity and function in some cancer patients.

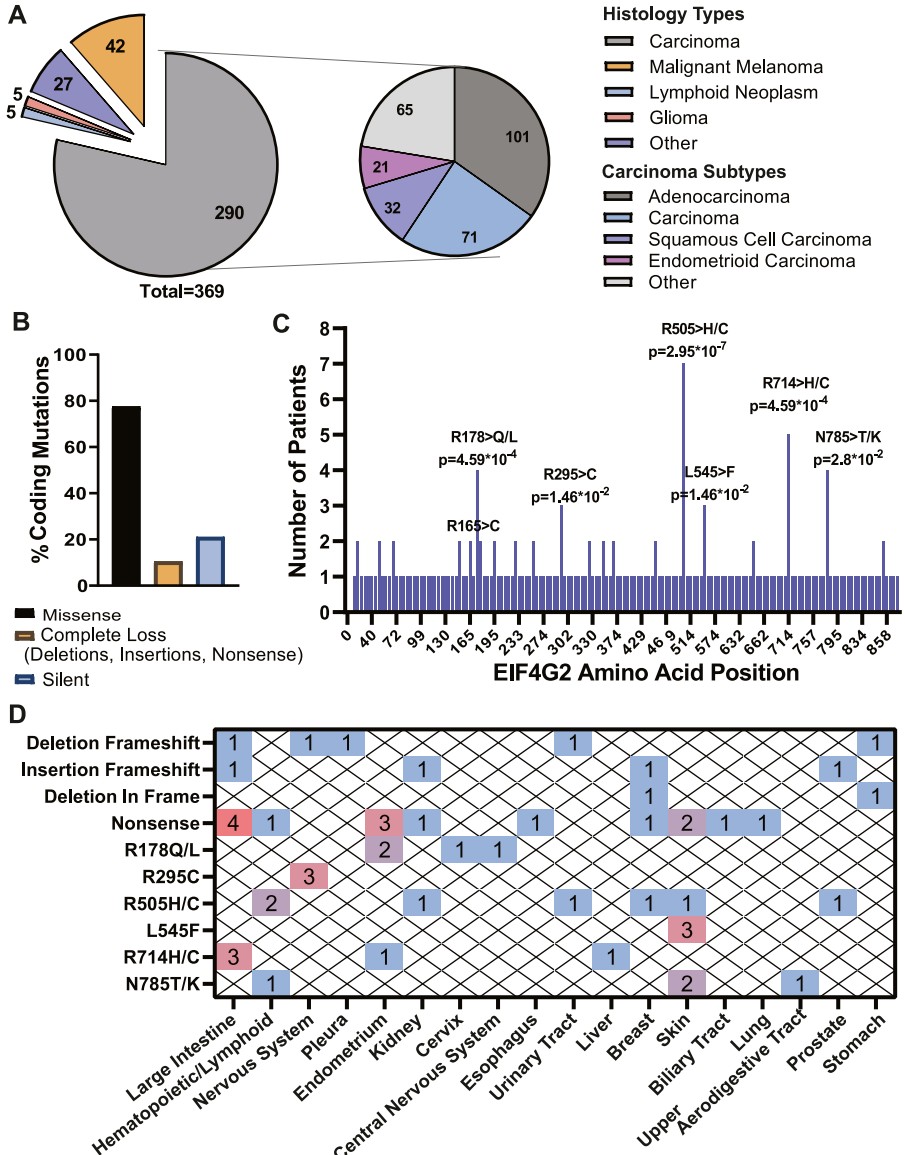

**Figure 1. Distribution of mutations in the *EIF4G2* gene in cancer patients.**
**(A)** Pie chart representing the distribution of 369 patient samples with *EIF4G2* mutations according to the tumor histology type (left). Pie-of-pie chart representing the tumor histology subtype of the 290 carcinoma patients with *EIF4G2* mutations (right). **(B)** Bar graph showing the mutation classification in 248 *EIF4G2* coding mutations. **(C)** Distribution and occurrence of 191 verified somatic missense mutations in the *EIF4G2* coding region, from 248 independent tumor samples, as identified in the COSMIC database. Positions with significant mutation occurrences are labeled with the amino acid substitution and *P*-value, calculated for windows of 1, 3, 9, 15, 30, and 60 bases, with steps of 1 (for windows of one base) or 3. The most significant *P*-values are shown, found with the one base window. **(D)** Number of patients with each deleterious or significant missense mutation, distributed according to the histology subtype.

## Point mutations in EIF4G2 functional domains alter its protein interactome

To dissect the potential effects that the missense mutations have on EIF4G2 function, the six significant hotspot missense somatic mutations were aligned to the domain organization scheme of the EIF4G2 protein (Fig 2A). Although 35% of the EIF4G2 protein is predicted to consist of unstructured regions with unknown functions (e.g., predicted EIF4G2 structure, AlphaFold Protein Structure Database; https://alphafold.ebi.ac.uk/entry/P78344), it also contains three distinct functional domains whose crystal structures have been determined at high resolution (23, 36, 37). These include the MIF4G domain (aa positions 78–308), the mainly α-helical MA3-like domain (positions 543–666), and the C-terminal W2 domain (720–907). Two of the patient-derived mutations localized to the MIF4G domain (R178 and R295) (Fig 2A),

which is known to interact with translation initiation factors EIF4A and EIF3, and also bind mRNA. As this implies a substantial contribution for this region to EIF4G2's potential tumor functions, we also included in the analysis an additional missense mutation (R165) that was previously shown to be important for RNA binding (36), despite the fact that it did not pass the significance score. The most abundant mutation position, R505, mapped to a large segment that is predicted to be unstructured with as-of-yet no known function. L545 is found at the start of the MA3-like domain, a region with unknown functions. It is mostly buried (37), and mutation to F, with a larger side chain, may affect the local folding. The R714 mutation is located C-terminal of the MA3-like domain, within the linker contacting it to the W2 domain (37). The N785 mutation mapped to the W2 domain that has been shown to interact with EIF5C, EIF2S2, and MNK1 (7, 23, 38, 39).

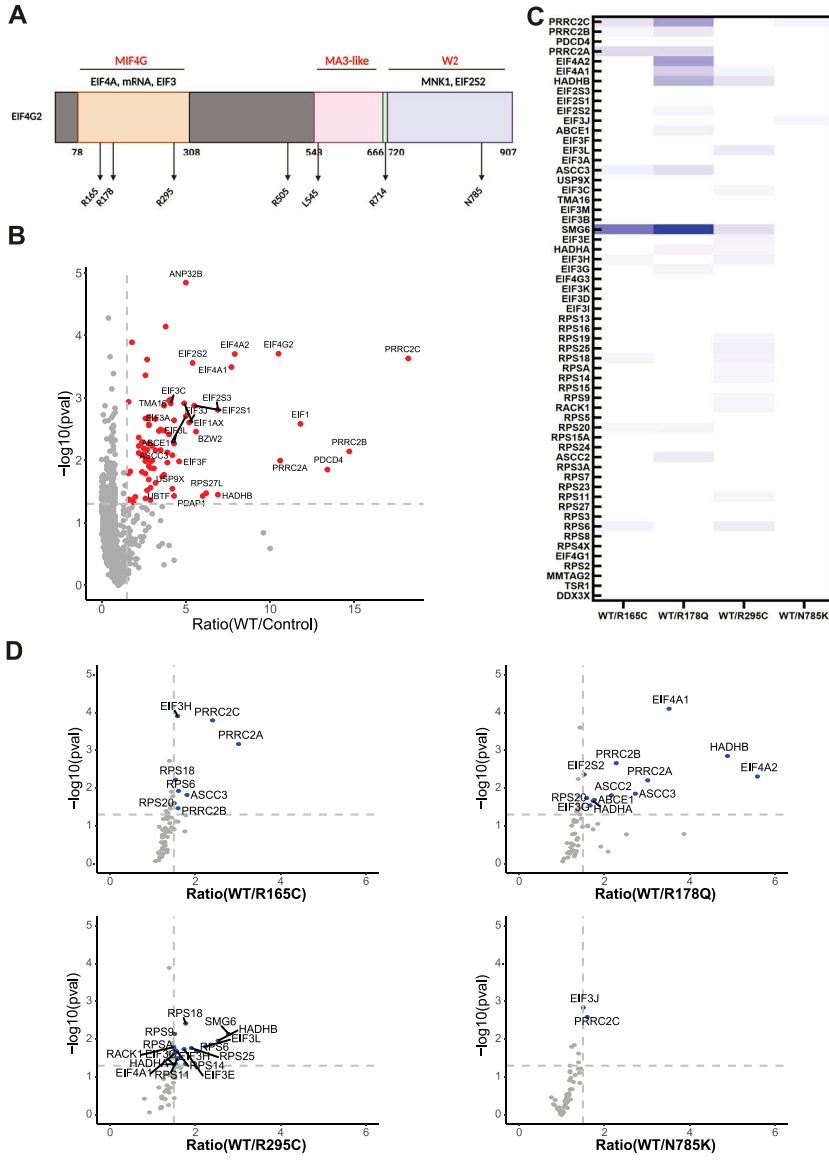

**Figure 2. MIF4G point mutations reduce EIF4G2 protein–protein interactions.**
**(A)** Schematic representation of the EIF4G2 protein. The domains with a known crystal structure are designated by amino acid position and labeled in red. Proteins shown to interact with these regions are indicated in black. The approximate locations of the significant mutations are represented by arrows in the relevant domains. Scheme was created with BioRender.com. **(B)** Volcano plot of the fold ratio of the abundance of the detected proteins in WT EIF4G2 versus control IP samples, versus their significance expressed as −log$_{10}$ P-value. Proteins with significantly increased abundance, that is, EIF4G2 interactors, are indicated in red. **(C)** Comparison of the binding abilities of EIF4G2 mutants with the 60 identified EIF4G2 interacting proteins. A heat map shows the fold ratio of the abundance of specific interacting proteins in the IP of the mutants compared with the WT EIF4G2 IP. The interactors presented are the ones identified as EIF4G2 interacting proteins (P < 0.05, fold change >1.5 of WT/control) regardless of their significance in the WT versus mutant comparison, and are listed in order of their relative abundance in the WT versus control IP. Decreased interaction with the mutant EIF4G2 is represented by a stronger blue strip; white indicates no change in protein abundance. Only significant fold changes >1.5 with a P-value < 0.05 after correction for levels of EIF4G2 in IP are indicated, and only mutants for which such changes were observed are shown. **(D)** Volcano plot of the fold-ratio protein abundance in IPs of WT EIF4G2 compared with either R165C, 178Q, R714C, or N785K mutants, versus their significance expressed as −log$_{10}$ P-value. Only interacting proteins are shown. Proteins showing significantly decreased abundance (>1.5 fold change, P < 0.05) in the mutant IP relative to the WT IP after correction for EIF4G2 levels are represented by blue dots.

Of these mutations, we chose to focus on the five that localized to the domains with established functions to investigate the mutations' outcome on EIF4G2's protein–protein interaction capability. In addition, we analyzed the outcome of the R505H substitution, which bore the most mutations, although it is found in an unstructured domain, anticipating that this may shed light on its unknown function. The individual point mutations, chosen based on their abundance in the patient samples (i.e., R165C, R178Q, R295C, R505H, R714H, and N785K), were constructed in FLAG-tagged EIF4G2 and transfected into HEK293T cells. Interactors were analyzed by co-immunoprecipitation (IP) followed by mass spectrometry (MS) analysis in two separate experiments, each comparing the protein interactome of WT EIF4G2 with either four or two of the mutants, respectively. EIF4G2 interactors were defined as those proteins that passed the threshold of mean abundance greater than 1.5 compared with the control IP, with a P-value < 0.05, in both MS runs. The first MS run yielded 83 interacting proteins (Fig 2B), whereas the second MS run yielded 151 (Fig S2A), with a shared set of 60 proteins; these 60 were defined as the set of EIF4G2 interactors. All but three have been previously linked to translation or mRNA regulation (Figs 2B and S2A and Table S1). 23 proteins were not previously reported by earlier studies of the protein interactome of EIF4G2 in mouse embryonic stem cells (40) and MDA-MB-231 breast cancer cells (15) (Fig S2B). They may represent indirect interactors, as unlike the study reported in reference 15, we did not include RNase nuclease in our IP studies. Consistent with these prior reports, the interactors mainly included components of the 43S pre-initiation complex (PIC), that is, canonical translation initiation factors of the EIF3, EIF4A, and EIF2S2 families and small ribosome subunits (20 and 24 proteins, respectively, Fig S2C). In addition, the data confirmed the strong interaction with the PRRC2 proteins, recently implicated in translation initiation, but whose functional connection to EIF4G2 is

not yet known (41, 42). Several of the newly identified interacting proteins have been more indirectly linked to translation initiation, including ribosome assembly factors (i.e., TMA16 and TSR1 (43, 44)), regulators of translation initiation (DDX3X (45), PDCD4 (46)), and several linked to mRNA surveillance and translation quality control, such as the endonuclease SMG6 that is involved in nonsense-mediated mRNA decay (47), ABCE1 of the no-go mRNA decay pathway (48), and ASCC2, ASCC3, and RACK1, which mediate the ribosome-associated quality control pathway (49, 50, 51) (Fig S2C, Table S1).

The R714H mutation near the W2 domain was the only mutation that did not significantly alter the repertoire of EIF4G2 interacting proteins in the complex (Table S1). The remaining mutants affected the interactome in different ways and to differing degrees, as evident by decreased relative abundance of specific interacting proteins in the IPs (>1.5 fold change, *P* < 0.05), after normalization to levels of the EIF4G2 protein (WT versus mutant) (Table S1, Fig 2C and D). The R295C mutation in the MIF4G mRNA binding domain affected the largest number of interacting proteins (17), including several EIF3 components and EIF4A1, which are known to bind to this domain. Yet, the effect was relatively modest, with the majority showing a less than twofold change in abundance, and none more than threefold. In contrast, R165C, also located in the MIF4G domain, showed a decreased interaction with nine proteins, including all PRRC2 family members and, most prominently, SMG6 (Fig 2C and D). The mutation did not impact on EIF4A, and affected binding to only one of the EIF3 subunits. The R178Q mutation led to decreased binding of 14 EIF4G2 interacting proteins (Fig 2C and D), with the greatest effect on the interaction with SMG6, HADHB, and all PRRC2 family members. There was also a prominent loss of binding to EIF4A1 and EIF4A2, which are critical for EIF4G2 function in translation initiation, and smaller declines in the interactions with other essential initiation factors, EIF3G and EIF2S2. The R505H mutant significantly affected the interaction with just three proteins (SMG6, TSR1, and RPS14), all of which showed modest declines in abundance in the IP, suggesting that this mutation does not contribute critically to EIF4G2's interactions (Fig S2D). Similarly, the C-terminal N785K mutant showed a significant but small decreased interaction with only PRRC2C and EIF3J (Fig 2C and D). Thus, the patient-derived mutations in the known EIF4G2 functional domains have an impact on EIF4G2's ability to interact with its protein partners in different ways, with the most prominent effects emerging from mutants within the MIF4G domain. Furthermore, they hint at possible binding domains for those interactors that have not yet been identified and/or mapped to the EIF4G2 structure.

### MIF4G domain mutants show differential effects on IRES- and uORF-mediated mRNA translation

To determine whether the changes in protein binding observed for the mutants had an effect on EIF4G2 functional activity, cellular translation assays were performed. Various established EIF4G2 targets that represent different translation initiation mechanisms were assessed. These included BCL2, representing IRES-directed targets (22, 26), and ROCK1 and WNK1, which contain cap-dependent uORFs that drive re-initiation of the downstream CDS in an EIF4G2-

dependent mechanism (14, 20). It should be noted that these proteins were chosen solely because of their confirmed status as EIF4G2 targets with well-established mechanisms of translation initiation, and their translation in the 293T model cell system does not necessarily imply that these are EIF4G2's relevant endogenous targets in cancer cells, where ultimately cell fate decisions will depend on the entire repertoire of EIF4G2 translation targets and the specific cellular context. We first confirmed the EIF4G2 dependency of all three proteins in EIF4G2 knock-out (KO) 293T cells, generated by the CRISPR/Cas9 system. Western blotting indicated that deletion of EIF4G2 resulted in reduced protein steady-state levels of endogenous BCL2, ROCK1, and WNK1 in 293T cells (Fig S3A).

The three targets were then tested as reporters in a cellular dual luciferase translation system. The reporter for BCL2 consisted of an A-cap structure followed by a hairpin loop added upstream of the BCL2 IRES to ensure cap-independent initiation of a firefly luciferase (F-LUC) gene (reference 7 and Fig 3A, scheme). For ROCK1 and WNK1 reporters, their entire 5'UTRs (including all uORFs) were inserted upstream of a Renilla luciferase (R-LUC) gene ((14), and schemes, Fig 3B and C). These were then co-transfected with either R-LUC or F-LUC, respectively, as internal controls, together with EIF4G2 WT or the panel of patient-derived mutants, in the EIF4G2 KO cells. The ability of the patient-derived mutants to drive translation of the reporters was compared with WT EIF4G2, after normalization to the second LUC control. The R714C mutation and N785K mutation at the C-terminal domain, and the R505H mutation in the unstructured domain, had no effect on any of the luciferase reporters compared with the WT (Figs 3A–C and S3B), indicating no functional significance for these translation initiation mechanisms. This was consistent with the MS analysis, in which only minor changes in the interactomes were evident for the R505H and the N785K mutants, and no changes for the R714C mutant. We cannot, however, at this point, exclude the possibility that these mutants may be impaired in other functions not tested here, such as EIF3D-mediated cap-dependent initiation or N6-methyladenosine–driven translation.

In contrast, the three MIF4G mutants all showed reduced activity toward the BCL2 IRES, but only one reduced the uORF-directed translation. Specifically, the R178Q mutant completely lost the ability to drive translation of the luciferase reporters for all targets tested; the LUC signals were reduced to the level of the control transfection (Fig 3A–C). The relative mRNA expression levels of the R-LUC and F-LUC reporters were assessed by quantitative PCR (qRT-PCR) to exclude decreased reporter transfection or gene expression as causes for the observed decreased reporter activity (Fig S3C and D). Thus, this mutant is incapable of driving either IRES-dependent or uORF-dependent initiation mechanisms. Notably, protein levels of the R178Q mutant were consistently lower than the WT and other mutants in all assays (Fig 3A–C). However, reducing the levels of WT EIF4G2 to comparable levels as the maximal R178Q expression by transfecting lower amounts of the WT EIF4G2 plasmid still did not equalize translation activity; EIF4G2 R178Q mutant exhibited a significantly lower translation activity toward the BCL2 reporter compared with WT EIF4G2, even when expressed at similar levels (Fig S3E and F). Thus, it is most likely that the loss of translation activity is a direct consequence of the mutation. Considering that the R178Q mutation had strong effects overall on the interactome and, specifically, reduced interactions with several essential

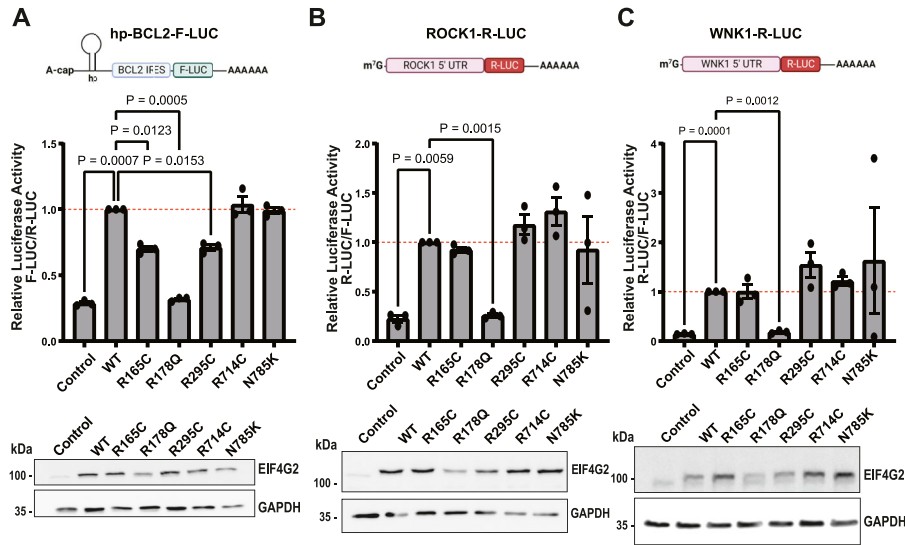

**Figure 3. Effect of EIF4G2 mutations on internal ribosome entry site (IRES)-dependent and uORF-dependent translation.**
**(A)** HEK293T EIF4G2 KO cells were co-transfected with the indicated EIF4G2 variants and a firefly luciferase (F-LUC) reporter driven by the BCL2 IRES and Renilla luciferase (R-LUC) reporter as an internal control. A schematic of the F-LUC reporter is shown, including a mutant A-cap structure and a hairpin upstream of the IRES sequence. F-LUC activity was quantified and normalized to the R-LUC activity; the graph shows the relative normalized LUC activity in all EIF4G2 transfectants with WT EIF4G2 transfection set as 1 (dashed red line). Total cell lysates were subjected to Western blot analysis using EIF4G2 and GAPDH antibodies as a loading control, shown below the graph. **(B, C)** HEK293T EIF4G2 KO cells were co-transfected with the indicated EIF4G2 variants and reporters containing ROCK1 5′UTR (B) or WNK1 5′UTR (C) upstream of R-LUC along with F-LUC as an internal control. Schematics of the R-LUC reporters are shown. R-LUC activity was quantified and normalized to the F-LUC activity; the graph shows the relative normalized LUC activity in all EIF4G2 transfectants with WT EIF4G2 transfection set as 1 (dashed red line). Total cell lysates were subjected to Western blot analysis using EIF4G2 and GAPDH antibodies as loading controls, shown below the graphs. Data information: for all panels, data are presented as individual data points and also as the mean ± SEM of three (A, C) or four (B) independent experiments, with a representative Western blot from one of the experiments shown. Significance was determined by matched one-way ANOVA followed by Dunnett's multiple comparison ad hoc test (comparing all variants with the WT EIF4G2 construct). Non-significant results ($P > 0.05$) were not indicated in the figure. Schemes were created with BioRender.com.
Source data are available for this figure.

canonical translation factors that are necessary for translation initiation, its loss of function is not surprising.

In contrast, the R165C and R295C mutants significantly impaired the ability to translate the F-LUC reporter bearing the BCL2 IRES by 30% relative to the WT EIF4G2 (Fig 3A), without impacting on the translation of the ROCK1 and WNK1 reporters (Fig 3B and C). Notably, Western blotting of lysates after co-transfection indicated that these mutants were expressed at similar or even greater levels compared with the WT EIF4G2, across the experiments (Fig 3A–C). These fluctuations inherent to the transfection conditions are not expected to affect translation activity, as calibration experiments with increasing concentrations of the transfected WT EIF4G2 plasmid indicated that reporter activity reached near-maximal activity at 1-μg plasmid, and even though EIF4G2 protein levels increased proportionally with further increases in plasmid trans-fected, reporter activity did not significantly increase (Fig S3E and F). Thus, we cannot attribute the decreased translation of the re-porters here to changes in the protein expression of the R165C and R295C mutants. Overall, the data imply a selective impact of these two mutations on IRES-mediated translation initiation but not translation directed by re-initiation from uORFs.

Each of the two mutants that selectively reduced IRES-dependent translation had different effects on the set of EIF4G2 interacting proteins, with four proteins showing reduced interac-tion for both: RPS6, RPS18, EIF3H, and SMG6. SMG6 was also affected by the R505H mutation, which retained its translation function toward the IRES-containing reporter; thus, loss of interaction with SMG6 alone does not explain the functional defect. As the remaining interactors, components of the 40S small ribosome subunit and EIF3, are necessary for translation initiation in general, it is not clear why loss of these interactions would specifically affect IRES-dependent translation but not uORF-dependent translation. It

is possible that IRES-dependent translation is particularly sensitive to minor changes in the assembly of the pre-initiation complex, more so than the mechanisms that mediate read-through of the uORF or re-initiation of the downstream main ORFs. Alternatively, as the MIF4G domain has been previously shown to mediate RNA binding, including, specifically, the R165 position, it is possible that these mutations specifically affect binding to target mRNAs bearing IRESes. In fact, EIF4G2 has been shown to be capable of directly binding mRNA targets with IRESes, such as p53 and HMGN3, in electrophoretic mobility shift assays (22, 28). It is reasonable to assume that binding to IRES-dependent targets differs in some inherent manner from binding to other targets, such as those that contain uORFs, and thus are more affected by the mutations at positions R165 and R295. Thus, these mutations serve as the first clue to potentially explain preferential target recognition, and further investigation is required to clarify these possibilities.

### 3D structural analysis predicts loss of protein interactions by MIF4G mutant

In order to understand why the mutations impact on EIF4G2's ability to bind its interacting proteins and promote translation, 3D structural analysis was performed. Model structures of the com-plete EIF4G2 generated by AlphaFold (https://alphafold.ebi.ac.uk/entry/P78344) or by RoseTTAFold (52) gave conflicting predictions as to the relative organization of the known domains and un-structured regions. We therefore focused on the experimentally solved structures, specifically the MIF4G domain in which the mutations that had functional consequences were located.

For the structural analysis of the EIF4G2 MIF4G domain and its interacting partner EIF4A, the structure of the human EIF4G2 MIF4G domain (36) was superimposed on the yeast EIF4G MIF4G-EIF4A (53)

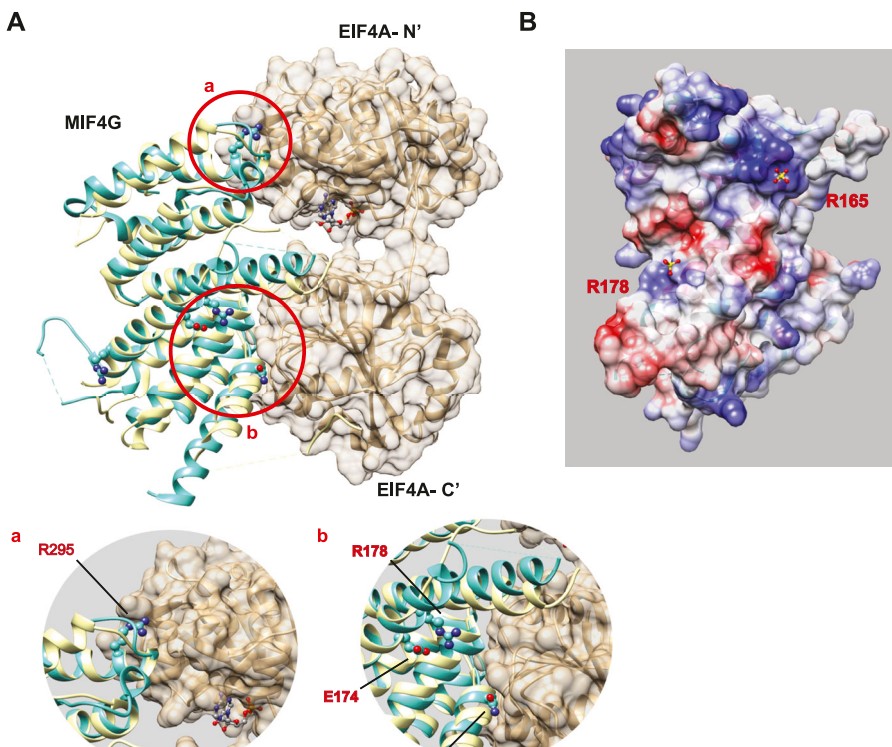

**Figure 4. EIF4G2 structural analysis predicts the possible outcome of patient-derived significant missense mutations.**

**(A)** Model of the interaction between the MIF4G domain of EIF4G2 and EIF4A based on the structure of the yeast complex between EIF4G (yellow) and EIF4A (beige) (PDB entry 2VSX). The structure of the human EIF4G2 MIF4G domain (cyan) (PDB entry 4IUL) was superposed on yeast EIF4G. As in the yeast complex, MIF4G interacts with two domains of EIF4A. **Inset a**, magnification of the circled area a, showing the interface with the N-terminal domain of EIF4A. R295 (K837 in yeast EIF4G) is marked in red. **Inset b**, magnification of the circled area b, showing the interface with the C-terminal domain of EIF4A, highlighting the position of N86 (N86 corresponds to N615 in yeast eIF4G). R178 (K709 in yeast) is shown, making an ion pair with E174 (E706 in yeast). **(B)** Surface of the DAP5 MIF4G domain (PDB entry 2VSX), showing a trough between R165 and R178. The surface is colored by the Coulombic potential, blue for positive, red for negative, and white for neutral. Sulfate ions are shown as ball-and-stick, yellow and red, respectively, for the sulfur and oxygen atoms. The sulfate ion near R165 is located within a deep and strongly positive cavity. The sulfate ion near R178 is located near a weakly positive surface region.

(Fig 4A). In the crystal structure, residue EIF4G K837, corresponding to human EIF4G2 R295, interacts electrostatically with the N-terminal domain of EIF4A (Fig 4A, inset a). Residue EIF4G N615, corresponding to human EIF4G2 N86, interacts with the C-terminal domain of EIF4A (Fig 4A, inset b); N86 is a well-studied position that is critical for EIF4A binding (7, 36). Both interactions are necessary to control the relative orientation of the two domains of EIF4A. The R295C mutation likely weakens the electrostatic interaction with EIF4A and possibly affects the relative position of the two domains of EIF4A. This can explain the moderate effects that this mutation had on the interaction with one of the EIF4A proteins. R165 is located near a deep cavity with a strong positive electrostatic charge, and serves as an mRNA binding site together with K108 and K112 (36) (Fig 4B). The R165C mutation may neutralize the positive electrostatic surface, and subsequently weaken the affinity for target mRNAs that rely on this region for binding, ultimately reducing the efficiency of their translation and the binding of proteins whose interactions are RNA-dependent.

The R178Q mutation had the most dramatic effects on the EIF4G2 protein, reducing its ability both to bind many interacting partners and to drive translation of all tested targets. These global loss-of-function effects most likely are related to the fact that its steady-state levels when ectopically expressed were vastly reduced compared with WT EIF4G2 (Fig S4A), which was particularly apparent when comparing protein expression upon transfection with a range of plasmid concentrations (Fig S3F). Mean mRNA levels were not significantly different from those of the WT, indicating this was not an issue with transfection of the plasmid or transcription of the mutant (Fig S4B). To explore the possibility that the point

mutation adversely affected protein stability, thereby leading to enhanced protein turnover, potential degradation mechanisms were examined. However, neither inhibition of ubiquitin-mediated degradation with the proteasome inhibitor bortezomib (Fig S4C) nor inhibition of autophagy/lysosomal-based degradation with the lysosome inhibitor hydroxychloroquine (Fig S4D) restored R178Q expression levels, implying that the reduced steady-state levels were not due to increased protein degradation by these pathways. We also excluded cleavage by specific proteolytic enzymes, as smaller EIF4G2 fragments were not observed on Western blotting (e.g., Fig S4C). Thus, it remains to be determined why this mutation affects its protein expression levels. As for possible effects on interactions, we note that although the positive charge of R178 is partially offset by the ion pairing with E174, the surrounding surface is moderately positive (Fig 4A). The mutation R178Q eliminates the positive side chain and precludes the ion pairing, resulting in neutralized surface potential. The structural analysis of the MIF4G domain alone did not provide an explanation for why the mutation would cause a major reduction in the steady-state levels or loss of functionality, but in the absence of an experimental structure of the entire protein, intra-molecular contacts and overall protein folding could not be assessed.

In conclusion, we have identified several EIF4G2 somatic mutations in primary tumors of cancer patients that show various impairments in binding to interacting proteins and the ability to direct mRNA translation. This correlation between impaired protein function and cancer is promising in establishing EIF4G2 as a potential tumor suppressor, at least in early pre-metastatic stages. Follow-up experiments specifically examining potential tumor-promoting

**Table 1. List of primers used for PCR cloning and qRT–PCR.**

| Primer | Sequence | Method |
|---|---|---|
| R165C_For | TGCCTCCTAATTTCCAAATTAC | cloning |
| R165C_Rev | TCTGAATGTGGTGCTTTG | cloning |
| R178Q_For | AAACTAGAAATGTTGATGTCT | cloning |
| R178Q_Rev | GGTTTTCAAATTCATCTTGT | cloning |
| R295C_For | TGTTTCCTGCTGCAGGATAC | cloning |
| R295C_Rev | AATCCTTGCTGGCAATTCC | cloning |
| R505H_For | ACACTCAAACACCACCTCTGG | cloning |
| R505H_Rev | GTGGTGGTTGTGCACTAGG | cloning |
| R714H_For | GATCAGAATAAGGACCACATGTTGGAG | cloning |
| R714H_Rev | CTCCAACATGTGGTCCTTATTCTGATC | cloning |
| N785K_For | CTAGTGAAGTAAAGCCCCCCAGCGATG | cloning |
| N785K_Rev | CATCGCTGGGGGGCTTTACTTCACTAG | cloning |
| FLAG-For | GACTACAAAGACGATGACG | qRT-PCR |
| FLAG-Rev | AACTCGCTGTTGCCAG | qRT-PCR |
| RLUC-For | GGTAAGTCCGGCAAGAGCGG | qRT-PCR |
| RLUC-Rev | GCCCCCCAGTCGTGGCCCAC | qRT-PCR |
| FLUC-For | CAGCCTACCGTGGTGTTCG | qRT-PCR |
| FLUC-Rev | GTGAGAACGTGTACATCG | qRT-PCR |
| HPRT-For | ATGGACAGGACTGAACGTCTT | qRT-PCR |
| HPRT-Rev | TCCAGCAGTCAGCAAAGAA | qRT-PCR |

attributes of these variants in cells and/or animal models are mandated to determine whether in fact loss of EIF4G2 function contributes to tumor development or growth.

# Materials and Methods

### Data collection and mutation analysis of the *EIF4G2* gene in human cancer

Whole-genome screen data from the Catalogue of Somatic Mutations in Cancer, COSMIC (http://cancer.sanger.ac.uk/cancergenome/projects/cosmic), version 96-38, were analyzed for somatic mutations of the *EIF4G2* gene in human cancer as previously described (35). In brief, unique independent samples were identified by comparing all their listed mutations. Confirmed somatic mutations of the *EIF4G2* gene were classified by mutation type. Further analysis of the coding region mutations calculated the significance of each cluster of mutations by statistics of observed versus expected mutations using a Poisson distribution for different nucleotide intervals.

### TCGA data mining

Mining of *EIF4G2* mRNA expression levels in healthy and primary tumor samples was done using the UCSC Xena Functional Genomic Explorer (54). All the chosen studies were TCGA-based, followed by search of genomic *EIF4G2* gene expression data, focusing on the sample_type phenotype.

### Cell lines and cell culture

HEK293T cells (ATCC CRL-3216) were cultured in DMEM (Biological Industries) with 10% FBS (Gibco), 1% penicillin–streptomycin (Biological Industries), and 1% L-glutamine (Biological Industries). Cells were routinely screened for mycoplasma. For the proteasome and lysosome inhibition assay, HEK293 EIF4G2 KO cells were transfected using Lipofectamine 2000 (Invitrogen) according to the manufacturer's recommendations, with either 5 μg empty pcDNA3, pcDNA3-FLAG-EIF4G2_WT, or pcDNA3-FLAG-EIF4G2_R178Q. 2 d after the transfection, cells were treated with 50 nM bortezomib (5043140001; Sigma-Aldrich) for 8 h or 10 μM hydroxychloroquine (90527; Sigma-Aldrich) for 24 h, or left untreated as a control.

### Structural analysis

Molecular graphics and structural analyses were performed with UCSF Chimera, developed by the Resource for Biocomputing, Visualization, and Informatics at the University of California, San Francisco, with support from NIH P41-GM103311 (55).

### Generation of EIF4G2 KO cells and EIF4G2 mutant plasmids

HEK293T EIF4G2 KO cells were generated by the CRISPR/Cas9 method, targeting exon 9 of the CDS. The following guides were cloned into a pKLV vector (kindly gifted by Y. Shaul, Weizmann Institute of Science, Israel): guide sequence ATTAGACCATGAACGAGCC with sense cloning primer, CACCATTAGACCATGAACGAGCC, and antisense cloning primer, TAAAACTGGCTCGTTCAGGTCT. 293T cells were transfected by standard calcium-phosphate transfection reagents. After 48 h, the transfected cells were treated with puromycin (Sigma-Aldrich) for selection of the transfected cells. The puromycin-resistant cells were transferred by limited dilution into a 96-well plate, then expanded and screened for the presence of the CRISPR KO by colony PCR followed by DNA sequencing using primers to exon 9 (forward primer: ATCAAGGAGCACATTCGGGC, reverse primer: GTGACAGGGAAGTTAGGCGA) followed by Western blot for validation.

EIF4G2 mutants were created from the WT FLAG-tagged EIF4G2 template in pcDNA3 (pcDNA3-FLAG-EIF4G2_WT (7)), using standard transfer-PCR or point mutagenesis cloning procedures with the primers listed in Table 1.

### Co-immunoprecipitation and mass spectrometry

HEK293T cells were transfected with 30 μg/15-cm plate of the following plasmids in two separate biological experiments by standard calcium-phosphate transfection: the first consisted of pcDNA3-FLAG-mCherry, pcDNA3-FLAG-EIF4G2_WT, pcDNA3-FLAG-EIF4G2_R165C, pcDNA3-FLAG-EIF4G2_R178Q, pcDNA3-FLAG-EIF4G2_R295C, or pcDNA3-FLAG-EIF4G2_R505H, and the second consisted of the FLAG-mCherry and EIF4G2 WT plasmids, pcDNA3-FLAG-EIF4G2_R714H, or pcDNA3-FLAG-EIF4G2_N785K. After 48 h, the cells were harvested and lysed in B-buffer (20 mM Hepes-KOH [pH 7.6], 100 mM KCl, 0.5 mM EDTA, 0.4% NP-40, and 20% glycerol) supplemented with protease and phosphatase inhibitors (Sigma-Aldrich). 5 mg lysates were incubated with FLAG beads (cat# A2220; Sigma-Aldrich) at 4°C for 2 h; a small amount was reserved for

Western blot validation of protein expression. The beads were washed three times with lysis buffer, and the bound protein was eluted using SDS buffer (5% SDS in 100 mM Tris–HCl, pH 7.4).

20 $\mu$g protein from the eluted samples of each experiment was subjected separately to in-solution tryptic digestion using the suspension trapping (S-trap micro-columns, ProtiFi) method as previously described (56). For the first IP experiment (control, WT EIF4G2, R165C, R178Q, R295C, and R505H mutants), the resulting peptides were loaded using split-less nano–Ultra Performance Liquid Chromatography (10 kpsi nanoACQUITY; Waters). The peptides were separated using an Aurora column (75 $\mu$m ID × 25 cm; IonOpticks) at 0.3 $\mu$l/min. Peptides were eluted from the column into the mass spectrometer using the following gradient: 2–30% B in 41 min, 30–90% B in 2 min, maintained at 90% for 3 min and then back to initial conditions. The nanoUPLC was coupled online to a quadrupole-Orbitrap mass spectrometer (timsTOF Pro; Bruker). Data were acquired in data-dependent acquisition with an ion mobility mode (data-dependent acquisition [DDA]–PASEF (57)), using a 1.1-s cycle-time method with 10 MS/MS scans. The ion mobility 1/K0 range was set to 0.60–1.60 Vs/cm$^2$, Energy Start in PASEF CID was set to 20.0 ev, and Energy End was set to 59.0 eV. Other parameters were kept as the default parameters of the DDA-PASEF method. The raw data were processed with FragPipe v17.1. The data were searched with the MSFragger search engine v3.4 against the human (Homo sapiens) protein database as downloaded from Uniprot.org, appended with common laboratory protein contaminants. Enzyme specificity was set to trypsin, and up to two missed cleavages were allowed. Fixed modification was set to carbamidomethylation of cysteines, and variable modification was set to oxidation of methionines and protein N-terminal acetylation. The quantitative comparisons were calculated using Perseus v1.6.0.7. Decoy hits were filtered out, and only proteins that had at least two valid values after logarithmic transformation in at least one experimental group were kept. For statistical calculations, missing values were replaced by random values from a normal distribution using the Imputation option in Perseus (width 0.3, downshift 1.8). Intensities were individually normalized to total protein or to the expression of EIF4G2 in each sample. A $t$ test of the logarithmic transformation was used to identify significant differences between the experimental groups, across the biological replica. Fold changes were calculated based on the ratio of geometric means of the different experimental groups.

For the second IP experiment (control, WT EIF4G2, R714H, and N785K mutants), each sample was loaded using split-less nano–Ultra Performance Liquid Chromatography as above, except that desalting of the samples was performed online using a reversed-phase Symmetry C18 trapping column (Waters) and the peptides were then separated using a T3 HSS nano-column (Waters) at 0.35 $\mu$l/min. Peptides were eluted from the column into the mass spectrometer using the following gradient: 4–27% B in 55 min, 27–90% B in 5 min, maintained at 90% for 5 min and then back to initial conditions. The nanoUPLC was coupled online through a nanoESI emitter (10 $\mu$m tip; New Objective; Woburn) to a quadrupole-Orbitrap mass spectrometer (Q Exactive HFX; Thermo Fisher Scientific) using a FlexIon nanospray apparatus (Proxeon). Data were acquired in a DDA mode, using a Top10 method. MS1

resolution was set to 120,000 (at 200 m/z); mass range, 375–1,650 m/z; AGC, 1 × 10$^6$; and maximum injection time, 50 ms. MS2 resolution was set to 15,000; quadrupole isolation, 1.7 m/z; AGC, 1 × 10$^5$; dynamic exclusion, 20 s; and maximum injection time, 60 ms. Raw data were processed with the MetaMorpheus algorithm, version 0.0.320 (58). The data were searched against the human (H. sapiens) protein database as downloaded from UniProt (www.uniprot.com), and appended with common laboratory protein contaminants. Enzyme specificity was set to trypsin, and up to two missed cleavages were allowed. Fixed modification was set to carbamidomethylation of cysteines, and variable modification was set to oxidation of methionines. Peptide and protein identifications were filtered at an FDR of 1%. The minimal peptide length was seven amino acids. Peptide identifications were propagated across samples using the match-between-runs option checked. Searches were performed with the label-free quantification option selected. The quantitative comparison and statistics were calculated as above.

EIF4G2 binding proteins were defined as those with abundance at least 1.5 orders of magnitude greater in the WT EIF4G2 IP compared with the control FLAG-Cherry IP sample, with $P < 0.05$, common to both IP and MS experiments. For these proteins, the relative abundance compared with the WT EIF4G2 IP was calculated for each EIF4G2 mutant IP; loss of binding was defined as at least a 1.5-fold decrease in the fold ratio of mutant versus WT EIF4G2 IPs, with $P < 0.05$, after correction for EIF4G2 levels in the IP.

## Western blot

Cells were lysed in RIPA lysis buffer (20 mM Tris, pH 8.5, 0.1% NP-40, 150 mM NaCl, 0.5% sodium deoxycholate, and 0.1% SDS) supplemented with 10 $\mu$l/ml 0.1 M PMSF (93482; Sigma-Aldrich) and 1% protease inhibitor (P8340; Sigma-Aldrich). Proteins were separated by SDS–PAGE and transferred to nitrocellulose membranes, which were incubated with the indicated antibodies: mouse anti-EIF4G2 (cat# 610742, RRID:AB_398065, 1:1,000 dilution; BD Biosciences), mouse anti-FLAG (cat# F3165, RRID:AB_259529, 1:1,000; Sigma-Aldrich), mouse anti-GAPDH (cat# MAB374, RRID:AB_2107445, 1:3,000; Millipore), rabbit anti-ROCK1 (cat# 4035, RRID:AB_2238679, 1:500; Cell Signaling Technology), rabbit anti-WNK1 (cat# 4979, RRID:AB_2216752, 1:500; Cell Signaling Technology) and mouse anti-tubulin (cat# T9026, RRID:AB_477593, 1:70,000; Sigma-Aldrich), and BCL2 (cat# sc-509, RRID:AB_626733, 1:1,000; Santa Cruz Biotechnology). Secondary antibodies consisted of either HRP-conjugated goat anti-mouse (cat# 115-035-003, RRID:AB_10015289; Jackson ImmunoResearch Labs) or anti-rabbit (cat# 111-165-144; Jackson ImmunoResearch), which were detected by enhanced chemiluminescence using EZ-ECL (Biological Industries).

## Luciferase translation assay

2 × 10$^5$ 293T EIF4G2 KO cells were seeded in six-well plates and transfected with 5 $\mu$g empty pcDNA3, pcDNA3-FLAG-EIF4G2_WT, pcDNA3-FLAG-EIF4G2_R165C, pcDNA3-FLAG-EIF4G2_R178Q, pcDNA3-FLAG-EIF4G2_R295C, pcDNA3-FLAG-EIF4G2_R505H, pcDNA3-FLAG-EIF4G2_R714H, or pcDNA3-FLAG-EIF4G2_N785K together with 1 $\mu$g pHP-IRES-BCL2-F-LUC plasmid (7, 22) with 1 $\mu$g pCIneo-R-LUC plasmid as a control, or alternatively with 1 $\mu$g pCIneo-WNK1-R-LUC or pCIneo-ROCK1-R-

LUC with 1 µg pEGFP-N3-F-LUC plasmids (kindly gifted from the Igreja Lab ([14]), Max Planck Institute, Germany), using Lipofectamine 2000 (Invitrogen) according to the manufacturer's recommendations.

48 h after transfection, cells were washed and harvested. The cell pellet was divided for luciferase activity, Western blot, and qRT-PCR analysis. For dual luciferase assay, cells were lysed and luciferase activity was measured on identical quantities of lysate according to the manufacturer's guidelines (E1960; Promega), using substrates for both F-LUC and R-LUC in sequential reactions. The luciferase signal was read using a Veritas microplate luminometer (Turner BioSystems). R-LUC/F-LUC or F-LUC/R-LUC activity of the control and mutants was normalized to that of WT EIF4G2.

### Reverse transcription and quantative real-time PCR (qRT-PCR)

0.5 µg RNA was mixed with 4 µl 5X reaction buffer and 1 µl RTase (AzuraQuant cDNA Synthesis Kit, cat# AZ1996; Azura Genomics). The reaction mix was incubated at 42°C for 30 min and denatured at 85°C for 10 min. The reaction was stopped by incubating the samples at 10°C for 10 min. qRT-PCR was performed using 166.6 ng cDNA, 10 µM forward and reverse primers (see Table 1), and 5 µl of AzuraView GreenFast qRT-PCR Blue Mix LR (AZ2305; Azura Genomics). Data were analyzed using QuantStudio 5 software (Thermo Fisher Scientific). mRNA levels were normalized to the housekeeping gene (HPRT), and the quantifications were calculated using the Livak method.

### Statistical analysis

Statistical analysis was done using GraphPad Prism 9. Data are presented as the mean values ± SEM of independent experiments. Matched one-way ANOVA with Dunnett's multiple comparison post hoc tests or paired two-tailed t tests were performed as indicated in the figure legends, with $P < 0.05$ considered statistically significant.

## Data Availability

The mass spectrometry proteomics data have been deposited in the MassIVE repository of the ProteomeXchange consortium (https://massive.ucsd.edu), with the dataset identifier MSV000092704.

## Supplementary Information

## Acknowledgements

We would like to thank Nadav Goldberg for his advice and many fruitful discussions and Lital Povodovski for preparing the HEK293T EIF4G2 KO cells. This research was supported by a grant from the Pasteur-Weizmann Council and from the Institut Pasteur and the Weizmann Institute of Science.

### Author Contributions

S Meril: conceptualization, data curation, formal analysis, validation, investigation, visualization, project administration, and writing—original draft, review, and editing.
M Bhalsen: validation, investigation, and writing—review and editing.
M Eisenstein: data curation, formal analysis, visualization, and writing—review and editing.
A Savidor: data curation and formal analysis.
Y Levin: data curation, formal analysis, and writing—review and editing.
S Bialik: formal analysis, investigation, visualization, and writing—original draft, review, and editing.
S Pietrokovski: data curation, formal analysis, and writing—review and editing.
A Kimchi: conceptualization, resources, supervision, funding acquisition, investigation, project administration, and writing—original draft, review, and editing.

### Conflict of Interest Statement

The authors declare that they have no conflict of interest.

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
