## [Reviewer comments · Life Science Alliance]

Life Science Alliance

Loss-of-function cancer-linked mutations in the EIF4G2 non-canonical translation initiation factor

Sara Meril, Marcela Bhalsen, Miriam Eisenstein, Alon Savidor, Yishai Levin, Shani Bialik, Shmuel Pietrokovski, and Adi Kimchi
DOI: <https://doi.org/10.26508/lsa.202302338>

Corresponding author(s): Adi Kimchi, Weizmann Institute of Science

Review Timeline:

Submission Date:	2023-08-24
Editorial Decision:	2023-10-11
Revision Received:	2023-11-15
Editorial Decision:	2023-12-07
Revision Received:	2023-12-12
Accepted:	2023-12-12

Transaction Report:

October 11, 2023

Re: Life Science Alliance manuscript #LSA-2023-02338-T

Prof. Adi Kimchi
Weizmann Institute of Science
Dept. of Molecular Genetics
Rehovot, rehovot 76100
Israel

Dear Dr. Kimchi,

Thank you for submitting your manuscript entitled "Loss-of-function cancer-associated mutations in the EIF4G2 non-canonical translation initiation factor" to Life Science Alliance. The manuscript was assessed by expert reviewers, whose comments are appended to this letter. We invite you to submit a revised manuscript addressing the Reviewer comments.

Thank you for this interesting contribution to Life Science Alliance. We are looking forward to receiving your revised manuscript.

Sincerely,

B. MANUSCRIPT ORGANIZATION AND FORMATTING:

Reviewer #1 (Comments to the Authors (Required)):

Comments to the authors

eIF4G2 is a ubiquitously expressed and abundant protein that drives the translation of selective mRNAs with roles in stem cell differentiation and proliferation, and embryonic development. As a non-canonical translation initiation factor, eIF4G2 aids in cap-dependent and independent-translation using alternative molecular mechanisms. Non-canonical initiation of translation is highly relevant under conditions in which cap-dependent translation is hindered or reduced. These include cellular stress, viral infections or mRNA features that limit the activity of the canonical translation initiation factors. Despite the importance of non-canonical translation in the control of protein synthesis, we still have limited information on the role of non-canonical translation initiation factors, such as eIF4G2, in disease onset and development.

Meril et al. address the contribution of eIF4G2 to cancer, which remains largely unknown. In this manuscript the authors analyze the presence of eIF4G2 somatic mutations in tumor samples of cancer patients using the COSMIC database. Functional analysis of missense mutations identified eIF4G2 variants with impaired protein interaction networks and translational activity. The results indicate eIF4G2 as a potential tumor suppressor protein, at least in colon and endometrial tumors. The data and experiments present in the manuscript are solid and clear and the authors have a large experience on the role of eIF4G2 in translation initiation. I do have some concerns regarding the presented data, that should be considered and addressed by the authors before publication.

Major points:

1. Following co-IP and MS analysis of Flag-tagged eIF4G2 variants transfected into HEK293T cells, the authors have analyzed the protein interaction network of WT and mutant proteins. Comparison of the respective interactomes, identified point mutations in eIF4G2 that significantly decreased the binding to specific proteins. However, detailed analysis of Table EV1, suggests that not all eIF4G2 variants used in the analysis were co-immunoprecipitated in similar amounts. Thus, some of the changes observed between WT and mutant proteins could just be a reflection of the differences in the amount of the bait/immunoprecipitated protein, as the authors comment in the case of the R178Q mutant. Can the authors provide the western blot figure of the immunoprecipitation experiment with the different eIF4G2 proteins analyzed by MS? And can the proteomic data be adjusted for differences in the amount of the bait protein? For instance, the authors claim that the "R295C mutation did not significantly altered the repertoire of the eIF4G2 interacting proteins". Yet, table EV1 indicates that the amount of mutant protein immunoprecipitated was almost 2-fold higher than the WT protein. In addition, residue R295 is involved in the interaction with the N-terminal domain of eIF4A. As discussed by the authors, the R295C substitution is expected to weaken the interaction with eIF4A. However, in Table EV1 there's no significant change in the amount of eIF4A bound to the R295C mutant. Can the authors include in the analysis a correction for protein amount (LFQ or iBAQ values taken into account for the analysis of the proteomic data)? Alternative, the authors could support their conclusions with co-IP assays in cells expressing similar levels of WT and mutant eIF4G2 proteins.
2. The expression levels of WT and mutant eIF4G2 proteins is also a concern in the reporter assays that determine the translational activity of the eIF4G2 variants. In the immunoblots presented in figure 3, and as pointed out in the text, not all eIF4G2 proteins have the same expression levels. To be able to claim that the changes in translational activity of the eIF4G2 variants result from alterations in binding to protein or RNA interactors, it is important to perform the reporter assays in conditions in which the eIF4G2 proteins are expressed at comparable levels. The authors could try to transfect higher (or lower) amounts of the plasmid DNAs encoding the different eIF4G2 variants to be able to compare their effects on the translational activity. This is particularly important for the R178Q which is consistently expressed at lower protein levels. In addition, if the authors want to support claims that this mutation interferes with protein stability, analysis of protein half-life could be performed.
3. The authors also discuss the possibility that some of the missense mutations could interfere with the binding of the eIF4G2 variant to RNA. Nevertheless, there is no experiment in this study that supports this hypothesis. Assays that compare the ability of the mutant proteins to bind to endogenous mRNA targets or to the BCL-2, WNK1 or ROCK1 reporter mRNAs are necessary to address this hypothesis and would improve the significance of this study.

Minor points:

1. The information regarding the R714H and N785K mutants is not present in Table EV1 to allow comparison of the different proteomes.
2. The authors have identified multiple eIF4G2 variants and focus on the point mutations observed on the domains of the protein. However, the most abundant variant, R505>H/C, was excluded from the analysis because it lies on an unstructured region of the protein. Since disorder regions in proteins also perform multiple regulatory and functional roles (binding platforms for proteins and nucleic acids, post-translational modification...), it would be interesting to at least investigate its contribution to eIF4G2 translational activity using the reporter assays described in the study.
3. The plasmid DNA concentration used in the transfection assays for the WT and mutant eIF4G2 proteins should be indicated in the materials and methods.
4. Can the authors show the differences in luciferase activity between the A-cap and the capped luciferase reporters? How efficiently is the A-cap reporter translated in the cells compared to the cap-dependent reporters? Can the authors exclude the possibility that poorly translated reporters (such as the A-cap reporter) are more prone to variation in their translation efficiency and therefore prone to small variations in the experimental conditions?
5. The material and methods should also include the information on the type of genomic change generated in the eIF4G2 locus following CRISPR-Cas9 genome editing.
6. The protein interactome of eIF4G2 was not analyzed in RNase-treated samples. It would be interesting to understand from the 23 new proteins identified in the eIF4G2 interactome how many are mediated by RNA. Can the authors comment on that?

Reviewer #2 (Comments to the Authors (Required)):

In this article, Meril and colleagues examine the effect of EIF4G2 somatic mutations on gene expression in the context of cancer. They mined the COSMIC database to identify somatic mutations found in cancer patients, and using proteomics and reporter assays, assess the effect of these mutations on the eIF4G2 protein interactome and its ability to modulate uORF- and IRES-mediated translation. Meril and colleagues provide a significant advancement towards the field of non-canonical translational control and its implications in diseased contexts. The paper is clearly written, and the experiments were expertly done. The conclusions are justified except for those noted below:

1. Screening of EIF4G2 mRNA Expression and Mutational Burden in Cancer Patients

The manuscript describes first the analysis of the Cosmic public cancer mutation database to identify somatic mutations in EIF4G2. They perform this analysis on a comprehensive scale, looking at 24 different cancer histology subtypes. Their analysis of mutation hotspots supports their conclusion that EIF4G2 has a possible biological importance in colon and endometrial tumours specifically, because mutations are more enriched in these subtypes.

2. Effect of EIF4G2 Somatic Mutations on the eIF4G2 Protein Interactome

Using mass spectrometry-based proteomics, the authors support their conclusion that the R178Q mutation induces the most significant change to eIF4G2's protein interactome, resulting in a loss of 21 interactors.

3. MIF4G Domain Mutations and their Effect on IRES- and uORF-mediated Translation

The authors conducted in vitro translation assays to explore the functional consequence of EIF4G2 somatic mutations on translation of known eIF4G2-sensitive mRNAs. There are some problems which need to be addressed. The authors conclude that the R178Q mutant was the only MIF4G mutant that resulted in the complete loss of ability to drive translation of all three targets tested. They note that the R178Q mutant protein levels were consistently lower than the WT and other mutants. In the most striking example of the BCL2 assay, it appears that the R178Q protein level is at least 2-fold lower than that of WT eIF4G2. It is therefore questionable whether the mutation is the cause of reduced, as it could be ascribed to lower protein stability or other causes. It would be very important to show protein quantifications for all the replicates. If transfections of the eIF4G2 mutants result in disparate expression patterns, they should repeat the experiment to ensure even expression to reach a strong conclusion.

4. Structural Analysis of MIF4G2 and W2 Domain Mutants

The authors use solved structures to rationalize the effect that their identified mutations may have on eIF4G2's protein interactome. Their interpretation is rational and clear.

Additional Comments:

1. In the conclusion the authors address the original observation that in many cancer subtypes, mutations in EIF4G2 lead to either loss of protein binding or its ability to mediate translation, which they state must implicate EIF4G2 as a potential tumor-suppressor. However, they ignore the fact that the three targets they chose to study (ROCK1, WNK1, and BCL2) have all been shown to be oncogenes and are involved in promoting tumorigenesis and/or metastasis. This contradicts the idea that loss of EIF4G2 impact the trajectory from normal to cancerous cells since mutated EIF4G2 would be incapable of facilitating the translation of oncogenic mRNAs. It is clearly more complex than this simple binary explanation. It would be worthy commenting

on this complexity in more detail.

2. The authors mention several times the PRRC2 proteins, which are relevant to their mass spectrometry findings. It would be helpful if they elaborated further on their function and the probable mechanism that mutated EIF4G2 would act through in relation to these proteins to affect translation.

In Figure 3 and Figure EV3A, the authors use different loading controls for their western blots. Is there a reason for this? It is customary (perhaps mandatory) to use the same housekeeping gene product as a loading control

We thank the reviewers for their favorable comments on the manuscript, and for the constructive critiques that have enhanced the quality of our data. We have addressed their concerns, as indicated below in red.

Reviewer #1

Major points:

1. A detailed analysis of Table EV1, suggests that not all eIF4G2 variants used in the analysis were co-immunoprecipitated in similar amounts. Thus, some of the changes observed between WT and mutant proteins could just be a reflection of the differences in the amount of the bait/immunoprecipitated protein, as the authors comment in the case of the R178Q mutant. Can the authors provide the western blot figure of the immunoprecipitation experiment with the different eIF4G2 proteins analyzed by MS? And can the proteomic data be adjusted for differences in the amount of the bait protein? For instance, the authors claim that the "R295C mutation did not significantly altered the repertoire of the eIF4G2 interacting proteins". Yet, table EV1 indicates that the amount of mutant protein immunoprecipitated was almost 2-fold higher than the WT protein.

In addition, residue R295 is involved in the interaction with the N-terminal domain of eIF4A. As discussed by the authors, the R295C substitution is expected to weaken the interaction with eIF4A. However, in Table EV1 there's no significant change in the amount of eIF4A bound to the R295C mutant. Can the authors include in the analysis a correction for protein amount (LFQ or iBAQ values taken into account for the analysis of the proteomic data)?

Alternative, the authors could support their conclusions with co-IP assays in cells expressing similar levels of WT and mutant eIF4G2 proteins.

The reviewer raised several related issues here that we have addressed:

a. Variations in IP levels of mutants: We did not have enough remaining protein to conduct westerns on the IP samples sent for MS analysis; based on our experience with transfectants of the mutants, as shown in Fig 3, their levels of expression, and enrichment after IP, will vary to a certain degree with each experiment, which is actually more accurately quantitated by the MS. We also believe that the MS quantitation is more accurate and statistically reliable than scanning and quantitation of co-IP-western blots. In the MS, the protein levels of each mutant and WT EIF4G2 varied, but for most, the mean difference was not significant ($p > 0.05$), except for the R295C mutant, which was IPed nearly twice as much as the WT (ratio WT/mutant, 0.56, p -val=0.022) (see Table S1, first 2 tabs, compare fold-change ratios of WT:mutant and p -values for EIF4G2, highlighted in yellow, across the mutants).

b. The normalization of MS data: As the reviewer recommended, we normalized the MS results of the interacting proteins to the levels of each IPed mutant, and the corrected ratios of abundance of each interacting protein in WT/mutant IPs are shown now in Table S1, third and fourth tabs. The actual intensities for each interacting protein, after correction for total protein amount by LFQ, are shown in the first two tabs of Table S1, along with the original ratios of abundance in WT vs mutant IPs

without normalization to EIF4G2 levels. The new analysis changed the effects of most of the mutations on the interacting proteins to different degrees, and the heat map in Fig 2C and the volcano plots in Fig 2D were updated accordingly, as was the text of the Results and Discussion (pg 10-11). Yet, other than R295C (see next comment), the overall conclusions of the analysis do not change: the MIF4G mutations have the most profound effects of the interactome, particularly the R178Q mutation, which decreased the abundance of a large number of interacting proteins to the greatest extent, while the mutations in regions outside of the MIF4G domain have very minor (N785K) to no effects (R714H) on the interactome.

c. Effects of R295C mutation on interactions: The largest changes in the new analysis were indeed observed for R295C, after normalization for its increased abundance, 17 proteins now pass the fold-change and *p*-value thresholds for decreased interactions (this is reflected in the updated Table S1, Fig 2C,D and pg 10 of the text). Thus, the 295 mutation does appear to affect the interactome, although many of these effects were relatively small. These include EIF4A1, which with a fold-change of 1.516 and a *p*-value of 0.024, just barely passed our thresholds. EIF4A2 did not show any significant differences in binding. Thus, there may indeed be a weakened interaction with EIF4A, but as other residues are also involved in the interaction with EIF4A, as explained in the text, this did not result in major changes in the interactions with the 2 EIF4A proteins. This contrasts with the R178C mutation, which strongly affected the interactions with both EIF4A1 and EIF4A2.

2. The expression levels of WT and mutant eIF4G2 proteins is also a concern in the reporter assays that determine the translational activity of the eIF4G2 variants. In the immunoblots presented in figure 3, and as pointed out in the text, not all eIF4G2 proteins have the same expression levels. To be able to claim that the changes in translational activity of the eIF4G2 variants result from alterations in binding to protein or RNA interactors, it is important to perform the reporter assays in conditions in which the eIF4G2 proteins are expressed at comparable levels. The authors could try to transfect higher (or lower) amounts of the plasmid DNAs encoding the different eIF4G2 variants to be able to compare their effects on the translational activity. This is particularly important for the R178Q which is consistently expressed at lower protein levels. In addition, if the authors want to support claims that this mutation interferes with protein stability, analysis of protein half-life could be performed.

For all but the R178Q mutant, transfection levels fluctuate between assays due to the inherent nature of such experiments co-expressing multiple plasmids (see the attached Source data for westerns of all experiments). In the revised manuscript, we added a calibration assay to assess the impact of such fluctuations on the reporter activity. Importantly, we found that even though protein levels increase proportionally with increasing transfected plasmid concentrations, the translation activity plateaus at 1 μ g plasmid, and does not increase with increased protein expression. Only the lowest 0.5 μ g plasmid concentration yields a corresponding translation activity that was significantly smaller than the higher concentrations (2.5 and 5 μ g plasmid), and even this was not proportional to the protein levels expressed. This important calibration control has now been added as the new Fig S3E,F (pg 13-14). Thus, fluctuations in

protein expression such as those observed with the different mutants are not expected to affect reporter activity, and importantly, the decreased translation activity observed for the R165C and R295C mutants towards the BCL2 IRES are unlikely to be attributed to any difference in protein expression.

As stated in the text, the R178Q mutant was consistently expressed to a lower degree than the others. Notably, increasing plasmid concentration in the transfections to the maximum did not achieve the levels of protein expression observed with WT EIF4G2, and in fact, at 5µg plasmid, expression levels were similar to those obtained with 1µg WT. Most importantly, when the reporter assays were compared at these equal expression levels, the R178Q mutant still maintained its minimal translation activity compared to the WT (see the new calibration western blot and Luc reporter assay comparing WT and R178Q at different plasmid concentrations, Fig S3E,F, pg 13). Thus, there is a strong correlation between R178Q's decreased functional activity and the reduction in binding to its protein interactors. To elucidate the mechanism behind the reduced expression of R178Q we have included additional experiments. Mean mRNA levels were not significantly different from that of the WT, indicating this was not due to issues with transfection of the plasmid or transcription/mRNA stability of the mutant (see new Figure S4B). Blocking lysosomal-based degradation pathways with HCQ or proteasome-mediated degradation with a proteasome inhibitor also did not rescue its expression (see new Figure S4C,D), and cleavage products were not observed on westerns (e.g. new Fig S4C), excluding protein degradation as a means for enhanced protein turn-over. While we suspect a folding/solubility issue, we could not perform a definitive assay to prove this hypothesis, and therefore leave the explanation for why R178Q is less efficiently expressed as an open question.

3. The authors also discuss the possibility that some of the missense mutations could interfere with the binding of the eIF4G2 variant to RNA. Nevertheless, there is no experiment in this study that supports this hypothesis. Assays that compare the ability of the mutant proteins to bind to endogenous mRNA targets or to the BCL-2, WNK1 or ROCK1 reporter mRNAs are necessary to address this hypothesis and would improve the significance of this study.

While we agree that this is an important point that would improve the study, we did not explore the mRNA binding of the EIF4G2 variants, but rely on previous data showing the importance of the EIF4G domain, in particular R165, for mRNA binding (ref 36). As the relevant endogenous mRNA targets of EIF4G2 in these cancers are not yet elucidated, and we do not know how the different mutants would behave towards different mRNAs, these experiments are beyond the scope of this work. We have rephrased this part of the discussion (pg 14) accordingly, also considering the newly analyzed effects on the interacting proteins.

Minor points:

1. The information regarding the R714H and N785K mutants is not present in Table EV1 to allow comparison of the different proteomes.

These 2 mutants were presented in the second tab of Table EV1 (now called Table S1) as they were analyzed in a separate MS experiment, as stated in the text.

2. The authors have identified multiple eIF4G2 variants and focus on the point mutations observed on the domains of the protein. However, the most abundant variant, R505>H/C, was excluded from the analysis because it lies on an unstructured region of the protein. Since disorder regions in proteins also perform multiple regulatory and functional roles (binding platforms for proteins and nucleic acids, post-translational modification...), it would be interesting to at least investigate its contribution to eIF4G2 translational activity using the reporter assays described in the study.

We agree that the mutation in the unstructured region can reveal functional information on these unknown domains, and therefore as suggested, we have added this analysis to our study. We found that this mutation had very minor effects on EIF4G2's interactions (3 proteins were decreased in abundance in its co-IP experiment: SMG6, TSR1 and RPS14 after normalization to EIF4G2 levels). Moreover, it did not affect EIF4G2's functional activity in the translation reporter assays. Thus, this mutation, although highly significant in the mutation cancer analysis, did not shed light on potential regulatory or functional roles of EIF4G2's unstructured domains, and thus we include this data as Supplementary material, see Table S1 for MS data (pg 11) and Supp Fig. S3B for the reporter translation assays (pg 12).

3. The plasmid DNA concentration used in the transfection assays for the WT and mutant eIF4G2 proteins should be indicated in the materials and methods.

We have added plasmid concentrations used for the IP-MS experiments to the Materials and Methods (pg 19); for luciferase assay transfections this was already provided (pg 23), or is indicated in the figure and its legends (i.e. new Fig S3E,F).

4. Can the authors show the differences in luciferase activity between the A-cap and the capped luciferase reporters? How efficiently is the A-cap reporter translated in the cells compared to the cap-dependent reporters? Can the authors exclude the possibility that poorly translated reporters (such as the A-cap reporter) are more prone to variation in their translation efficiency and therefore prone to small variations in the experimental conditions?

All raw data used for the graphs shown in Fig 3 are presented in the source data, indicating the absolute translation levels of all reporters. A-capped reporters are used in this case in order to ensure IRES-dependent translation only; without an IRES to direct translation initiation independently of the cap, the A-cap would block translation altogether. We expect IRES-dependent translation to be inefficient compared to cap-dependent translation. For that matter, the presence of uORFs also suppresses translation of the main ORFs that follow; the uORF is translated in a cap-dependent manner as usual, and the main ORF is only translated following read-

through of the uORF or re-initiation after its translation is terminated, which are far less efficient processes. As has been demonstrated by multiple groups in the established literature, it is the specific function of EIF4G2 to mediate these less efficient modes of translation. Thus, all DAP5-dependent reporters are expected to be translated in a relatively inefficient manner. For this reason, reporter activity was normalized to the activity obtained with WT EIF4G2, so that the low activity is inherently controlled for within the experiment. We do not believe that the IRES reporter is more prone to small changes in experimental conditions, as the remaining mutants tested (R714C, R505H and N785K) did not affect reporter activity despite similar experimental conditions and variability.

5. The material and methods should also include the information on the type of genomic change generated in the eIF4G2 locus following CRISPR-Cas9 genome editing.

This information, describing the targeted locus and guides used for targeting, and sequencing of the affected locus after KO, has been added to the Materials and Methods (pg 18-19).

6. The protein interactome of eIF4G2 was not analyzed in RNase-treated samples. It would be interesting to understand from the 23 new proteins identified in the eIF4G2 interactome how many are mediated by RNA. Can the authors comment on that?

We did not examine the RNA-dependence of the EIF4G2 interactome. It is possible that some of the 23 newly identified interactors are dependent on RNA for binding, a possible explanation for why they were not previously observed in at least one of the published IP-MS experiments that were done in the presence of RNase. This point has been added to the text (pg 9). However, while this is an interesting point, an analysis of the new interactors is beyond the scope of this paper.

Reviewer #2 (Comments to the Authors (Required)):

Major Comment

The authors conducted in vitro translation assays to explore the functional consequence of EIF4G2 somatic mutations on translation of known eIF4G2-sensitive mRNAs. There are some problems which need to be addressed. The authors conclude that the R178Q mutant was the only MIF4G mutant that resulted in the complete loss of ability to drive translation of all three targets tested. They note that the R178Q mutant protein levels were consistently lower than the WT and other mutants. In the most striking example of the BCL2 assay, it appears that the R178Q protein level is at least 2-fold lower than that of WT eIF4G2. It is therefore questionable whether the

mutation is the cause of reduced, as it could be ascribed to lower protein stability or other causes. It would be very important to show protein quantifications for all the replicates. If transfections of the eIF4G2 mutants result in disparate expression patterns, they should repeat the experiment to ensure even expression to reach a strong conclusion.

We added new calibration experiments to address this important issue. We achieved equal expression for the 178 mutant by reducing plasmid concentrations of the WT 5-fold, and under these conditions, the loss of the translation activity of the R178Q mutant was still prominent, as compared to the WT EIF4G2 (new Fig S3E,F). Thus, as we explain in the discussion, we currently suggest that the loss in translation activity of R178Q mutant is due mostly to the functional effects of the mutation (i.e., on its protein interactome). Importantly, the activity of the WT EIF4G2 did not vary significantly with increased protein expression (new Fig S3E,F), showing that fluctuations in expression of the protein should not alone affect translation activity. Expression levels of the EIF4G2 variants are shown for all experiments in the Source Data; the inherent fluctuations in the co-expression experiments, and the presence of variable non-specific bands in the vicinity of EIF4G2 which added noise to the quantification, did not allow for quantification across experiments, as the variability precluded statistical significance in the general comparison. Despite the fluctuations for the remaining variants, the translation activity was reasonably consistent across the experiments, as predicted from the concentration curve analysis that we performed with WT EIF4G2.

Additional Comments:

1. In the conclusion the authors address the original observation that in many cancer subtypes, mutations in EIF4G2 lead to either loss of protein binding or its ability to mediate translation, which they state must implicate EIF4G2 as a potential tumor-suppressor. However, they ignore the fact that the three targets they chose to study (ROCK1, WNK1, and BCL2) have all been shown to be oncogenes and are involved in promoting tumorigenesis and/or metastasis. This contradicts the idea that loss of EIF4G2 impacts the trajectory from normal to cancerous cells since mutated EIF4G2 would be incapable of facilitating the translation of oncogenic mRNAs. It is clearly more complex than this simple binary explanation. It would be worthy commenting on this complexity in more detail.

It should be noted that analysis of the translation efficiency of these 3 reporter mRNAs in 293 cells was used as a model system for translation assays and does not necessarily reflect the true targets in cancer cells. These reporters were chosen because they have been clearly demonstrated by multiple groups to be targets of EIF4G2 translated by the different mechanisms, either IRES for BCL-2 or uORFs for ROCK1 and WNK1. Their use in these assays does not prove that they are the relevant targets of EIF4G2 in cancer. EIF4G2 has many protein translation targets,

some of which have opposing functions (e.g. anti-apoptosis BCL-2, IAPs and pro-apoptosis APAF-1 and MYC). Moreover, work by the Schneider group and our own work (submitted, see BioRxiv preprint server, <https://doi.org/10.1101/2023.09.14.557672>) indicate that there are many targets that are relevant to cancer progression or suppression, depending on the cancer type, and that it is the complete repertoire of targets within the specific cellular context that will determine the role that EIF4G2 plays in cancer. We have clarified this point on pg 11.

2. The authors mention several times the PRRC2 proteins, which are relevant to their mass spectrometry findings. It would be helpful if they elaborated further on their function and the probable mechanism that mutated EIF4G2 would act through in relation to these proteins to affect translation.

There is not much known about the PRRC2 proteins and their role in translation. PRRC2A and PRRC2B were originally shown to be M6A readers that affect mRNA stability. Two recent papers have directly connected the family members to translation initiation (e.g. Ref # 41, 42). While the strong interaction with EIF4G2 is highly suggestive of a shared role in translation, and Ref 41 presents some data supporting this hypothesis, it is not known how, and or even fully proven whether, EIF4G2 and PRRC2 proteins act together to mediate translation initiation. In fact, there is poor overlap between the reported potential mRNA targets of the two proteins in the current literature. Therefore, we cannot at this point speculate as to how the mutations may affect any such cooperative function. This point has been now stressed in the text, pg 10.

In Figure 3 and Figure EV3A, the authors use different loading controls for their western blots. Is there a reason for this? It is customary (perhaps mandatory) to use the same housekeeping gene product as a loading control

We customarily simultaneously run 2 loading controls in our experiments. We have replaced the blots shown in Fig 3B so that the same one is shown for all westerns. The blot of the loading control shown for ROCK1 in EV3A (now Fig S3A) was actually mislabeled and is indeed tubulin, as shown for the other blots.

December 7, 2023

RE: Life Science Alliance Manuscript #LSA-2023-02338-TR

Prof. Adi Kimchi
Weizmann Institute of Science
Dept. of Molecular Genetics
Rehovot, rehovot 76100
Israel

Dear Dr. Kimchi,

Thank you for submitting your revised manuscript entitled "Loss-of-function cancer-linked mutations in the EIF4G2 non-canonical translation initiation factor". We would be happy to publish your paper in Life Science Alliance pending final revisions necessary to meet our formatting guidelines.

- please add a Category for your manuscript in our system
- please update your callouts for the Supplementary Figures in the manuscript Fig S1A, B
- the 3 uploaded files marked as Related Manuscript Files, appear to be Source Data. If this is correct, please be sure to label them as such.

A. FINAL FILES:

B. MANUSCRIPT ORGANIZATION AND FORMATTING:

Sincerely,

Reviewer #1 (Comments to the Authors (Required)):

The authors have considered most of my suggestions and questions. I appreciate the efforts to include additional controls in their experiments and to try to elucidate several of the raised criticisms. The revised manuscript is therefore an improved version of this study. I fully support the publication of the revised version of the manuscript.

Reviewer #2 (Comments to the Authors (Required)):

In this article, Meril and colleagues examine the effect of EIF4G2 somatic mutations on gene expression in the context of cancer. They mined the COSMIC database to identify somatic mutations found in cancer patients, and using proteomics and reporter assays, assess the effect of these mutations on the eIF4G2 protein interactome and its ability to modulate uORF- and IRES-mediated translation. The paper provides a significant advance in the field of non-canonical translational control and its implications in diseased contexts. The authors addressed the main concern shared by the reviewers regarding normalization levels within the reporter assays by conducting calibration assays to rule out some of the questions of how the varying protein expression levels might affect translation activity in their experimental system. Considering the significant revisions made by the authors the paper warrant publication in Life Science Alliance.

December 12, 2023

RE: Life Science Alliance Manuscript #LSA-2023-02338-TRR

Prof. Adi Kimchi
Weizmann Institute of Science
Dept. of Molecular Genetics
Rehovot, rehovot 76100
Israel

Dear Dr. Kimchi,

Thank you for submitting your Research Article entitled "Loss-of-function cancer-linked mutations in the EIF4G2 non-canonical translation initiation factor". It is a pleasure to let you know that your manuscript is now accepted for publication in Life Science Alliance. Congratulations on this interesting work.

DISTRIBUTION OF MATERIALS:

Again, congratulations on a very nice paper. I hope you found the review process to be constructive and are pleased with how the manuscript was handled editorially. We look forward to future exciting submissions from your lab.

Sincerely,
